# CONTINUAL LEARNING VIA NEURAL PRUNING

## ABSTRACT

We introduce Continual Learning via Neural Pruning (CLNP), a new method aimed at lifelong learning in fixed capacity models based on neuronal model sparsification. In this method, subsequent tasks are trained using the inactive neurons and filters of the sparsified network and cause zero deterioration to the performance of previous tasks. In order to deal with the possible compromise between model sparsity and performance, we formalize and incorporate the concept of *graceful forgetting*: the idea that it is preferable to suffer a small amount of forgetting in a controlled manner if it helps regain network capacity and prevents uncontrolled loss of performance during the training of future tasks. CLNP also provides simple continual learning diagnostic tools in terms of the number of free neurons left for the training of future tasks as well as the number of neurons that are being reused. In particular, we see in experiments that CLNP verifies and automatically takes advantage of the fact that the features of earlier layers are more transferable. We show empirically that CLNP leads to significantly improved results over current weight elasticity based methods. CLNP can also be applied in single-head architectures for completely fixed capacity continual learning.

## 1 INTRODUCTION

Continual learning, the ability of models to learn to solve new tasks beyond what has previously been trained, has garnered much attention from the machine learning community in recent years. This is driven in part by the practical advantages promised by continual learning schemes such as improved performance on subsequent tasks as well as a more efficient use of resources in machines with memory constraints. There is also great interest in continual learning from a longer-term perspective, in that any approach towards artificial general intelligence needs to be able to continuously build on top of prior experiences.

As it stands today, the main obstacle in the path of effective continual learning is the problem of catastrophic forgetting: machines trained on new problems forget about the tasks that they were previously trained on. There are multiple approaches which seek to alleviate this problem, ranging from employing networks with many submodules Rusu et al. (2016); Fernando et al. (2017); Lee et al. (2017) to methods which penalize changing the weights of the network that are deemed important for previous tasks Kirkpatrick et al. (2016); Zenke et al. (2017); Hou & Kwok (2018). These approaches either require specialized genetic algorithm training schemes or still suffer catastrophic forgetting, albeit at a smaller rate. In particular, to the best of our knowledge, there is no form of guarantee regarding the performance of previous tasks among the fixed capacity models which use standard SGD training schemes.

In this work we introduce a simple continual learning algorithm for fixed capacity networks which can be trained using standard gradient descent methods and by construction suffers *zero* deterioration on previously learned problems during the training of new tasks. We take advantage of the over-parametrization of neural networks by using an activation based neural pruning scheme to train models which only use a fraction of their width. We then train subsequent tasks utilizing the unused capacity of the model. By cutting off certain connection in the network, we make sure that new tasks can take advantage of previously learned features but cause no interference in the pathways of the previously learned tasks.

MAIN CONTRIBUTIONS

- We introduce Continual Learning via Neural Pruning (CLNP) with the following properties:

- Given a network with activation based sparsity, CLNP trains new tasks utilizing the unused weights of the network in a manner which prevents any catastrophic forgetting of the previous tasks while taking advantage of the already learned features.
    - CLNP provides simple diagnostics in the form of the number of remaining and reused neurons and filters. In particular, in experiments we see that CLNP automatically takes advantage of the fact that the features of earlier layers are more transferable.
- We formalize the idea of graceful forgetting, the notion that it is preferable to suffer from forgetting in a controlled manner if it helps regain network capacity and prevents uncontrolled loss of performance. We use this idea to balance network sparsity and model accuracy.

- We show empirically that using an activation based neural pruning sparsification scheme, we outperform previous approaches based on weight elasticity. We also demonstrate in one example that using a slightly more advanced variation of our sparsification method, the network suffers virtually no loss of performance, either from catastrophic forgetting or from sparsification.

- In its single-head incarnation, CLNP provides one of the first viable single-head continual learning algorithms, and the first to avoid catastrophic forgetting by construction as far as we are aware.

## 2    RELATED WORK

**Lifelong learning.** Prior work addressing catastrophic forgetting generally fall under two categories. In the first category, the model is comprised of many individual modules at each layer and forgetting is prevented either by routing the data through different modules Fernando et al. (2017) or by successively adding new modules for each new task Rusu et al. (2016); Lee et al. (2017). This approach often has the advantage of suffering zero forgetting, however, the structure of these networks is specialized. In the case of Rusu et al. (2016); Lee et al. (2017) the model architecture is not fixed, and in the case of Fernando et al. (2017) training is done using a tournament selection genetic algorithm. In contrast, algorithms in the second category do not require any specific network structures or training schemes. Here, forgetting is minimized by penalizing changes of weights which are deemed important Kirkpatrick et al. (2016); Zenke et al. (2017); Ritter et al. (2018); Hou & Kwok (2018). These approaches, generally referred to as weight elasticity methods, have the advantage of simpler training schemes but still suffer catastrophic forgetting, albeit at a smaller rate than unconstrained training. Some approaches combine aspects from both categories, for instance Schwarz et al. (2018) utilizes weight elasticity methods to aid knowledge distillation into a fixed capacity knowledge base.

The approach we take in this paper falls under the second category with standard network structure and training scheme, and forgetting is prevented by constraining certain weights from changing. However, it shares the main advantage of the first category approaches in that we suffer zero catastrophic forgetting during the training of subsequent tasks. In particular, our method can be thought of as following the spirit of the path based approach of Fernando et al. (2017) using activation based sparsification and the idea of graceful forgetting. Since our method is directly comparable to other second category approaches, we will provide quantitative comparisons with other methods in this category.

**Network superposition.** Our algorithm can be thought of as a non-destructive way of superposing multiple instances of the same architecture using sparsification. Prior work in this direction (e.g. Saxena & Verbeek (2016); Shazeer et al. (2017)) are primarily concerned with neural architecture search and are not aimed at preventing forgetting. The most relevant work in this area is Cheung et al. (2019), which can be thought of as an approximate Fourier space implementation of our approach.

**Sparsification.** While sparsification is a crucial tool that we use, it is not in itself a focus of this work. For accessibility, we use a simple neuron/filter based sparsification scheme which can be thought of as a single iteration variation of Hu et al. (2016).

## 3    METHODOLOGY

The core idea of our method is to take advantage of the fact that neural networks are generally over-parametrized (Neyshabur et al., 2018). A manifestation of this over-parametrization is through the practice of sparsification, i.e. the compression of neural network with relatively little loss of performance (LeCun et al., 1989; Sun et al., 2015; Gale et al., 2019). As an example, Luo et al. (2017)

show that VGG-16 (Simonyan & Zisserman, 2014) can be compressed by more than 16 times without significant loss of performance. In this section we first show that given an activation based sparse network, we can leverage the unused capacity of the model to develop a continual learning scheme which suffers no catastrophic forgetting. We then discuss the idea of graceful forgetting to address the tension between sparsification and model performance in the context of lifelong learning. In the next section, we will see empirically that even fairly small networks are sufficiently over-parametrized for our method to provide competitive results.

It is important to differentiate activation based neuronal sparsity from parameter based weight sparsity. The former implies only a subset of the neurons or filters of each layer assume non-zero values on a given dataset whereas the latter means that many of the weights are zero but does not necessarily imply the existence of permanently inactive neurons. In the remainder of the paper, when we mention sparsity, we are referring to activation based neuronal sparsity.

In what follows we will discuss sparsity for fully connected layers by looking at the individual neurons. The same argument goes through identically for individual channels of convolutional layers.

## 3.1 GENERALITIES

Let us assume that we have a trained network which is sparse in the sense that only a subset of the neurons of the network are active. Networks with this form of sparsity can be thought of as narrower networks embedded inside the original structure. There are many approaches that aim to train such sparse networks with little loss of performance (e.g. Luo et al. (2017); Hu et al. (2016)). We will discuss our sparsification method in detail in §3.2.

Fig. 1 shows a cartoon of our approach, where we have a network with activation based neuronal sparsity, where the active and inactive neurons are respectively denoted by blue and grey nodes. Based on the connectivity structure, the weights of the network can also be split into three classes. First, denoted in blue in Fig. 2, we have the active weights $W^{\text{act}}$ which connect active nodes to active nodes. Next we have the weights which connect any node to inactive nodes, we call these the free weights $W^{\text{free}}$, denoted in grey. Finally we have the weights which connect the inactive nodes to the active nodes, we call these the interference weights $W^{\text{int}}$, denoted in red dashed lines. A more precise definition of the active and inactive neurons and weights is given in §3.2.

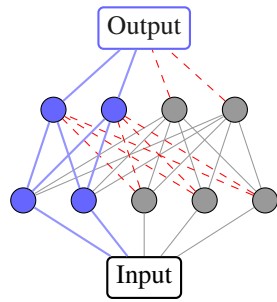

Figure 1: The partition of a network with neuronal sparsity into active (blue), inactive (grey) and interference (red) parts.

The crux of our approach is the simple observation that if all the interference weights $W^{\text{int}}$ are set to zero, the free weights $W^{\text{free}}$ can be changed arbitrarily without causing any change whatsoever to the output of the network. We can therefore utilize these weights to train new tasks without causing any harm to the performance of the previous tasks.

We can further split the free weights into two groups. First, the weights which connect active nodes to inactive nodes. These are the weights that take advantage of previously learned features and are therefore responsible for transfer learning throughout the network. We also have the weights that connect inactive nodes to inactive nodes. These weights can form completely new pathways to the input and train new features. A complimentary diagnostic for the amount of transfer learning taking place is the number of new active neurons at each layer after the training of subsequent tasks. Given that an efficient sparse training scheme would not need to relearn the features that are already present in the network, the number of new neurons grown at each stage of training is an indicator of the sufficiency of the already learned features for the purposes of the new task. We will see more of this point in §4.

**Output architecture.** In order to fully flesh out a continual learning scheme, we need to specify the connectivity structure of the output nodes. Starting with a network trained on the first task with the interference weights severed (Fig. 2 middle), there are two intuitive routes that we can take. In order to train a new task, one option is to use a new output layer (i.e. a new head) while saving the previous output layer. This option, demonstrated in Fig. 2 on the left, is known as the multi-head approach and

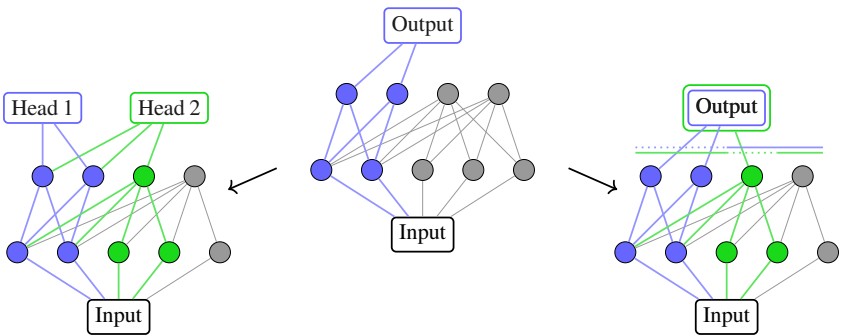

Figure 2: Middle: A network with neuronal sparsity. Left, right: the multi-head and single-head expansions.

is standard in continual learning. Because each new output layer comes with its own sets of weights which connect to the final hidden layer neurons, this method is not a fully fixed capacity method. In our approach to continual learning, training a multi-head network with a fully depleted core structure, i.e. a network where are no more free neurons left, is equivalent to final layer transfer learning.

In scenarios where the output layer of the different tasks are structurally compatible, for example when all tasks are classification with the same number of classes, we can use a single-head approach. Demonstrated in Fig. 2 on the right, in this approach we use the same output layer for all tasks, but for each task, we mask out the neurons of the final hidden layer that were trained on other tasks. In the case of Fig. 2, only green nodes in the final hidden layer are connected to the output for the second task and only blue nodes for the first task. This is equivalent to a dynamic partitioning of the final hidden layer into multiple unequal sized parts, one part for each task. In practice this is done using a multiplicative masking operation with a task dependent mask, denoted in Fig. 2 by dashed lines after the final hidden layer. This structure is truly fixed capacity, but is more restrictive to train than its multi-head counterpart. CLNP is among the first methods which provides a viable algorithm for single-head continual learning, and the first to avoid catastrophic forgetting by construction as far as we are aware.

## 3.2 Methodology details

**Sparsification.** So far in this section we have shown that given a sparse network trained on a number of tasks, we can train the network on new tasks without suffering any catastrophic forgetting. We now discuss the specific sparsification scheme that we use throughout this paper, which is similar in spirit to the network trimming approach put forward in Hu et al. (2016).

Our sparsification method is comprised of three parts. First, during the training of each task, we add an $L^1$ weight regularizer to promote sparsity in the network and to regulate the magnitude of the weights of the network. The coefficient $\alpha$ of this regulator is a hyperparameter of our approach. Also, since different layers have different weight distributions, we can gain more control over the amount of sparsity in each layer by choosing a different $\alpha$ for each layer. The second part of our sparsification scheme is post-training neuron pruning based on the average activity of each neuron. The third part of the scheme is referred to as fine-tuning and involves adjusting the surviving weights of the network after pruning and is done by retraining the network for a few epochs while only updating the weights which survive sparsification. This causes the model to regain some of the performance loss suffered because of pruning. To achieve a yet higher level of sparsity, one can iterate the pruning and fine-tuning steps multiple times. In our method, we use one or two iterations of training, with or without fine tuning depending on the task.

In §3.1, we partitioned the network into active and inactive parts. A precise definition of these different partitions is as follows. Given network $N$, comprised of $L$ layers, we denote the neurons of each layer as $N_l$ with $l = 1, \cdots, L$. Let us also assume that the network $N$ has been trained on dataset $S$. In order to find the active and inactive neurons of the network, we compute the average activity over the entire dataset $S$ for each individual neuron. We identify the active neurons $N_l^{\text{act}}$, i.e. the blue nodes in Fig. 2, as those whose average activation magnitude exceeds some threshold parameter $\theta$: $N_l^{\text{act}} = \{N_l \,|\, \mathbb{E}_S(|N_l|) > \theta\}$. The inactive neurons are taken as the complement $N_l^{\text{inact}} = N_l \setminus N_l^{\text{act}}$. The

threshold value $\theta$ is a post-training hyperparameter of our approach. Similar to the $L^1$ weight regulator hyperparameter $\alpha$, $\theta$ can take different values for the different layers. Furthermore, if $\theta = 0$, $N_l^{\text{inact}}$ would be given by the neurons in the network which are completely dead and the function being computed by the network is entirely captured in $N_l^{\text{act}}$. We can therefore view $N_l^{\text{act}}$ as a compression of the network into a sub-network of smaller width. Based on their connectivity structure, the weights of each layer are again divided into active, free and interference parts, respectively corresponding to the blue, grey and red lines in Fig. 2.

**Summary.** The overall algorithm of our approach in its multi-head incarnation is given in Alg. 1. During the training of each task, all interference weights are first set to zero and the free weights are re-initialized. We then place the new head for the $i$'th task and train while updating only the free weights $W^{\text{free}}$. We then update the set of active neurons to include the newly activated neurons during the training of the $i$'th task and accordingly update the set of interference and free weights. We then repeat the process for task $i+1$.

### 3.3 GRACEFUL FORGETTING

While sparsity is crucial in our approach for the training of later tasks, care needs to be taken so as not to overly sparsify and thereby reduce the model's performance. In practice, model sparsity has the same relationship with generalization as other regularization schemes. As sparsity increases, initially the generalization performance of the model improves. However, as we push our sparsity knobs (i.e. the $L^1$ regulator and activity threshold) higher and make the network sparser, eventually both training and validation accuracy will suffer and the network fails to fit the data properly. This means that in choosing these hyperparameters, we have to make a compromise between model performance and remaining network capacity for future tasks.

---

**Algorithm 1:** Continual Learning via Neural Pruning (CLNP) - Multi-head

**Data:** datasets $S = \{S_i\}$,
network with $l$'th layer neurons $N_l$
$N_l^{\text{act}}, W^{\text{int}} \leftarrow \emptyset$;
$W^{\text{free}} \longleftarrow W(N_l \rightarrow N_{l+1})$;
**for** $S_i \in S$ **do**
    $W^{\text{int}} \longleftarrow 0$;
    Re-initialize $W^{\text{free}}$;
    Place new head $N_L^{(i)}$;
    Train on $S_i$ updating only $W^{\text{free}}$;
    Choose highest $\theta$ such that Val. Acc
    $N_l^{\text{act}} \longleftarrow N_l^{\text{act}} \cup \{N_l \mid \mathbb{E}_{S_i}(|N_l|) > \theta\}$;
    $N_l^{\text{inact}} \longleftarrow N_l \setminus N_l^{(i)}$;
    $W^{\text{int}} \longleftarrow W(N_l^{\text{inact}} \rightarrow N_{l+1}^{(1)})$;
    $W^{\text{free}} \longleftarrow W(N_l \rightarrow N_{l+1}^{\text{inact}})$;

---

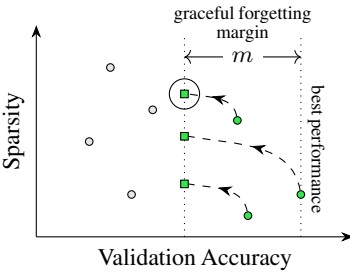

Figure 3: Cartoon of sparsification hyperparameter search. Each circular node corresponds to a model trained with a different pretraining hyperparameter. The green circles are the models within $m\%$ of the best validation accuracy. The dashed lines and green squares denote the trajectory and final end-point of the validation/sparsity of the model as the pruning threshold $\theta$ is increased. The model with the highest sparsity is then chosen.

This brings us to a subject which is often overlooked in lifelong learning literature generally referred to as graceful forgetting (Weng & Luciw, 2009; Weng & Luciw, 2010). This is the general notion that it would be preferable to sacrifice a small amount of accuracy in a controlled manner, if it reduces catastrophic forgetting of this task and helps in the training of future tasks. We believe any successful fixed capacity continual learning algorithm needs to implement some form of graceful forgetting scheme. In our approach, graceful forgetting is implemented through the sparsity vs. performance compromise. In other words, after the training of each task, we sparsify the model up to some acceptable level of performance loss in a controlled manner. We then move on to subsequent tasks knowing that the model no longer suffers any further deterioration from training future tasks. This has to be contrasted with other weight elasticity approaches which use soft constraints on the weights of the network and cannot guarantee future performance of previously trained tasks. We will see in the next section that using the exact same network structure, our approach leads to noticeably improved results over existing methods.

The cartoon of the choice of sparsity hyperparameters depicted in Fig. 3. We first scan over a range of hyperparameters (i.e. $\alpha$, the $L^1$ weight regulator and $\xi$, the learning rate) using grid search and

note the value of the best validation accuracy across all hyperparameters.[1] We then pick the models which achieve validation accuracy within a margin of $m\%$ of this best validation accuracy. The margin parameter $m$ controls how much we are willing to compromise on accuracy to regain capacity and in experiments we take it to be generally in the range of $0.05\%$ to $2\%$ depending on the task. We sparsify the picked models using the highest activation threshold $\theta$ such that the model remains within this margin of the best validation accuracy.[2] We finally pick the hyperparameters which give the highest sparsity among these models. In this way, we efficiently find the hyperparameters which afford the highest sparsity model with validation accuracy within $m\%$ of its highest value.

After pruning away the unused weights and neurons of the model with the hyperparameters chosen as above, we report the test accuracy of the sparsified network. This algorithm for training and hyperparameter grid search does not incur any significant additional computational burden over standard practice. The hyperparameter search is performed in standard fashion, and the additional steps of selecting networks within the acceptable margin, scanning the threshold, and selecting the highest sparsity network only require evaluation and do not include any additional network training.

# 4 EXPERIMENTS

We evaluate our approach for continual learning on the permuted MNIST (LeCun & Cortes, 2010) and split versions of CIFAR-10 and CIFAR-100 (Krizhevsky et al., 2009), and compare to previous results. In all experiments we use Adam optimizer with hyperparameters as in Kingma & Ba (2014). For network initialization, as well as reinitializing the free weights, we employ He-normalization (He et al., 2015), using all neurons (not just free ones) to compute the fan in.

As mentioned in the previous section, by construction, the performance of previous tasks does not degrade whatsoever when training subsequent tasks. We therefore do not include performance degradation as a function of number of subsequent task plots. In exchange, we provide the final performance plots after all tasks have been trained.

## 4.1 PERMUTED MNIST

In this experiment, we look at the performance of our approach on ten tasks derived from the MNIST dataset via ten random permutations of the pixels. To make direct comparison with prior work, we employ the most restrictive network reported in a recent survey on this benchmark by Swaroop et al. (2019): a 2 layer network with 100 neurons in each layer, ReLU activation (Glorot et al., 2011) and a softmax multi-class cross-entropy loss trained using batches of 256 samples.

We use a learning rate of $0.002$ and $L^1$ weight regularization $\alpha = 10^{-7}, 10^{-5}, 10^{-6}$ respectively for the first, second and final layers. Finally, we sparsify using 2 iterations of pruning and retraining with a graceful forgetting margin of $m = 3\%$. This is the smallest margin that allows for the training of all 10 tasks without running out of capacity in the body of the network, after which the performance of the model on newly learned tasks (e.g. task 11) would be lower by around 10%. After training the final task, the average test accuracy of the network across all tasks is $95.8 \pm 0.15\%$ (mean $\pm$ standard deviation across 5 trials). The exhaustive comparison table from Swaroop et al. (2019), updated with our results, is reproduced in Tab. 1 in the supplementary materials for convenience. The best prior performance on this task using the same narrow network architecture is provided by Ritter et al. (2018) who achieve[3] a training accuracy of $\sim 96\%$. However, this method requires the computation of the (block-diagonal Kronecker factored) Hessian and is therefore much higher complexity than CLNP. The fact that CLNP on this narrow network matches state-of-the-art performance is an indication that excessively wide networks are not necessary for CLNP to be competitive.

To verify the viability of CLNP in a single-head architecture, we also perform the experiment with a wider 2-layer network of width 2000. This network (or rather its multi-head variant) was used in earlier

---

[1] The choice of both $\alpha$ and $\xi$ affects the sparsity of the trained network. In practice, however, this dependence is easily captured via the same grid search used to determine best model performance and does not invoke any extra computational cost.

[2] More specifically, we ramp up $\theta$ one layer at a time starting from the deepest layer. In practice, we find that the choice of direction does not make a large difference in the final sparsity of the model.

[3] Value inferred from a graph in the source paper.

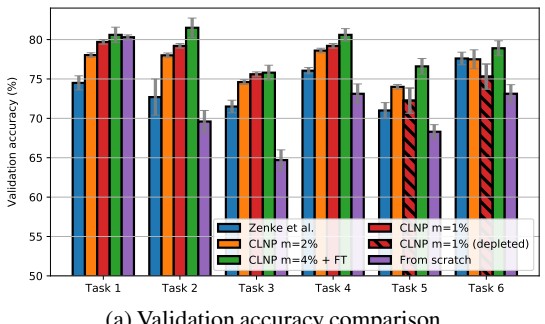

(a) Validation accuracy comparison

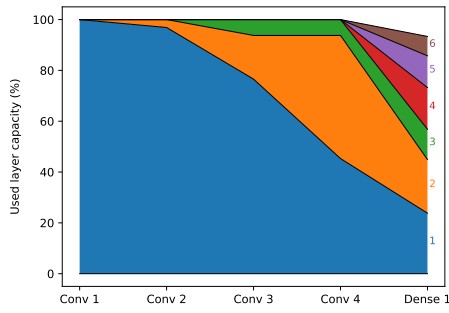

(b) Average network capacity usage per task

Figure 4: CIFAR-10 and split CIFAR-100 results on multi-head network. (a) Final validation accuracy after all tasks are trained. The red dashed bars on task 5 and 6 denote that the network ran out of capacity after training task 4. At this point the performance of the network suffers as can be seen in comparison to other methods.(b) Percentage of the capacity of each layer used after the training of each task.

experiments on this benchmark (e.g. Zenke et al. (2017)) and provides an interesting comparison with current and previous multi-head algorithms. Here, we achieve an average test accuracy across 10 tasks of $98.42\pm0.04\%$, which compares favorably to the $97\%$ average performance achieved by Zenke et al. (2017). While the single-head algorithm requires a wide final layer, as can be seen in Tab. 1, it provides very competitive results on such networks.

## 4.2 SPLIT CIFAR-10/CIFAR-100

In this experiment, we train an image classifier sequentially, first on CIFAR-10 (task 1) and then on CIFAR-100 split into 10 different tasks, each with 10 classes (tasks 2-11). This experiment is a better benchmark of continual learning since it tests how well an algorithm implements transfer learning across different tasks. However, because of the higher computation requirements of this task, there are not many comparison points for this benchmark. To facilitate comparison with previous algorithms, we use the same multi-head network as in Zenke et al. (2017). We also use a wider single-head network for comparison. The structure of these two networks are given respectively given in Tab. 1 and Tab. 2 of the supplementary materials. We train sequentially on the first 6 tasks of this problem using Adam optimizer with learning rate 0.001 and an $L^1$ weight regularizer coefficient $\alpha = 5 \times 10^{-5}$. We perform the experiment under three different CLNP schemes: first using a single iteration of pruning with graceful forgetting margins of $m = 1\%$ and $m = 2\%$ without fine-tuning. We then repeated the experiment under a third scheme defined by a larger margin of $m = 4\%$ but this time followed by fine-tuning. In each case, we perform the experiment 5 times and report the validation accuracy and its standard deviation on the 6 tasks.

The results of the experiment are shown in Fig. 4a. Here, the most ambitious scheme with a graceful forgetting of less than $m = 1\%$, runs out of capacity after the fourth task is trained. We notice that after the model capacity is depleted, the performance of this scheme plummets, showing the necessity for the presence of free neurons in the core of the network during training. The more moderate forgetting scheme $m = 2\%$, however, maintains high performance throughout all tasks and does not run out of capacity until the final task is learned. The best performance, however, comes from $m = 4\%$ graceful forgetting scheme followed by one iteration of fine-tuning. We see that here there is virtually no catastrophic forgetting on the first task (if anything the model performs even better after pruning and retraining as has been reported in previous sparsity literature LeCun et al. (1989); Hu et al. (2016)). The remaining tasks also get a significant boost from this improved sparsification method. In this case, CLNP beats prior results by Zenke et al. (2017) by more than 4% on average.

Although we outperform previous methods, the narrow structure of this network is ill-suited for our algorithm. An ideal network for our method would have an evenly over-parametrized capacity at all layers of the structure but with this network, 95% of the parameters of the entire network are concentrated in a single layer (Krizhevsky, 2014). Fig. 4b shows the network capacity usage per task for the moderate $m = 1\%$ forgetting scheme averaged over the 5 runs. The first task alone almost fills up the entirety of the first and second convolutional layer capacities, leaving little room for new

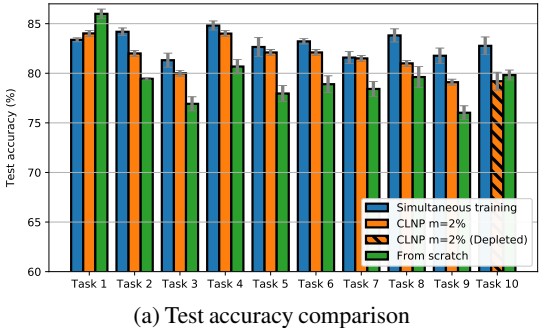

(a) Test accuracy comparison

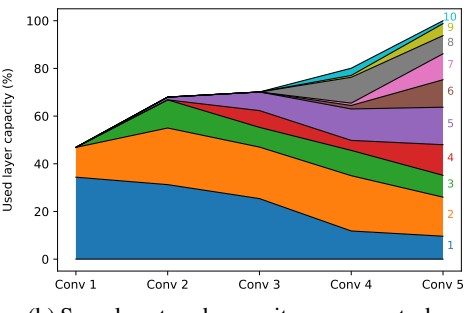

(b) Sample network capacity usage per task

Figure 5: CIFAR-10 and split CIFAR-100 results on wide single-head network. (a) Final test accuracy after all tasks are trained. The orange dashed bar on task 10 denotes that the network ran out of capacity after training task 9. At this point the performance of the network suffers as can be seen in comparison to other methods. (b) Percentage of the capacity of each layer used after the training of each task. Lower layers stop training new channels after training a few tasks. Deeper layers keep training new channels for all tasks.

convolutional channels to be learned from future tasks. The fact that even with this structure we still outperform previous methods is an indication that a large amount of transfer learning is taking place.

In what follows, we look at two variations on our approach to this problem: first with a slightly more advanced sparsification scheme and second with a single-head network of much wider width.

**Wide single-head network.** To gain further insight into the interplay of transfer learning and network width in networks with activation based neuronal sparsity, we train a much wider fully convolutional network on the same task (Tab. 2 of supplementary materials). The use of this very wide network is made possible only by the computational simplicity of our algorithm and would prove prohibitive for competing methods which require, for example, the computation of the Hessian. This time we use a single-head structure making the task more challenging because of the fixed total capacity of the network. In this case, we use a slightly different training scheme, employing a heldout validation set comprised of 10% of the training samples for hyperparameter selection and early stopping. We run the experiment 5 times and report the mean and standard deviation of the test accuracy after all tasks have been trained.

The wider network in this experiment has the capacity to train 10 tasks (i.e. CIFAR-10 plus 9 out of the 10 CIFAR-100 tasks) before depleting its final hidden layer. Unlike multi-head structures where a depleted final hidden layer leads to naive transfer learning on new tasks, for single-head structures, a depleted final hidden layer simply has no free connections to the output and the network is entirely fixed. Fig. 5a shows the results of this experiment.[4] For comparison we have provided the results of multi-task training and training each task individually from scratch. On average, CLNP outperforms training from scratch we outperform by a margin of $2-5\%$ and trails simultaneous training by $1-3\%$. This trend is reversed only on the first task (CIFAR-10) which has an abundance of training samples and does not benefit from being trained concurrently with the other tasks. Details of the multi-task training is given in the supplementary materials section.

The average neuron capacity usage per task is given in Fig. 5b. There are a number of interesting features in this graph. First, tasks number 1-7 each roughly takes up the same number of neurons of the final hidden layer. This is again as expected since the neurons of this layer are connected to the output layer and are not reused in order to avoid interference. By the time we get to task 8, more than 85% of the capacity of this layer is depleted. This directly affects the performance of the network which can be seen in terms of reduced test accuracy both in comparison to the results from multi-task training and individual training from scratch.

---

[4] The only comparable single-head experiment on this dataset that we are aware of was done by Cheung et al. (2019) who use a single-head 6-layer convolutional network. Using superposition techniques they find that the first task suffers about a 10% catastrophic forgetting (from $\sim 72\%$ down to $\sim 63\%$ after training 4 new tasks, accuracy on other tasks not reported). In comparison our method suffers less than a 2% drop via graceful forgetting and remains unchanged during the 9 subsequent tasks.

The most interesting observation in the training of the wide network is in the number of new channels learned at each layer for each consecutive tasks. The first convolutional layer trains new channels only for task 1 and 2. The second and third convolutional layers grow new channels up to task 3 and task 5 respectively. The fourth layer keeps training new channels up to the last task. The fact that the first layer grows no new channels after the second task implies that the features learned during the training of the first two tasks are fully utilized and deemed sufficient for the training of the subsequent tasks. The fact that this sufficiency happens after training more tasks for layers 2 and 3 is a verification of the fact that features learned in lower layers are more general and thus more transferable in comparison with the features of the higher layers which are known to specialize Yosinski et al. (2014). This observation implies that models which hope to be effective at continual learning need to be wider in the higher layers to accommodate this lack of transferability of the features at these scales.

## 5    CONCLUSION

In this work we have introduced an intuitive lifelong learning method which leverages the over-parametrization of neural networks to train new tasks in the inactive neurons/filters of the network without suffering any catastrophic forgetting in the previously trained tasks. We implemented a controlled way of graceful forgetting by sacrificing some accuracy at the end of the training of each task in order to regain network capacity for training new tasks. We showed empirically that this method leads to results which exceed or match the current state-of-the-art while being less computationally intensive. Because of this, we can employ larger models than otherwise possible, given fixed computational resources.

Our methodology comes with simple diagnostics based on the number of free neurons left for the training of new tasks. Model capacity usage graphs are informative regarding the transferability and sufficiency of the features of different layers. Using such graphs, we have verified the notion that the features learned in earlier layers are more transferable. We can leverage these diagnostic tools to pinpoint any layers that run out of capacity prematurely, and resolve these bottlenecks in the network by increasing the number of neurons in these layers when moving on to the next task. In this way, our method can expand to accommodate more tasks and compensate for sub-optimal network width choices.

Our algorithm is crucially dependent on the sparsification method used. In this work we employed a neuron activity sparsification scheme based on a single pruning iteration. We demonstrated that using a slightly more advanced sparsification scheme, i.e. adding a fine tuning step after pruning can lead to even better results. This shows that our results, can only improve with more effective sparsification methods. This observation opens the path for a new class of neuronal sparsity focused lifelong learning methods as an alternative to current weight elasticity based approaches.

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

## A    PERMUTATION MNIST RESULT COMPARISON

For convenience, here we reproduce Tab. 2 from Swaroop et al. (2019) for comparison with prior work. We update this table with a new result which was not available at the time this comparison was published. With the exception of our single-head experiment, all other methods use a multi-head architecture.

| Method/Paper | Hidden layer size | Number of tasks | Final average test accuracy |
|---|---|---|---|
| CLNP (ours) | {100, 100} | 10 | 95.8±0.15% |
| CLNP single-head (ours) | {2000, 2000} | 10 | 98.42±0.04% |
| VCL | {100, 100} | 10 | 93±1% |
| *VCL + 200 random coreset | {100, 100} | 10 | 94.6±0.3% |
| Previous VCL | {100, 100} | 10 | 90% |
| *Previous VCL + 200 random coreset | {100, 100} | 10 | 93% |
| Kronecker-factored Laplace | {100, 100} | 50 | 90% |
| Kronecker-factored Laplace | {100, 100} | 10 | 96%** |
| EWC | {2000, 2000} | 10 | 97% |
| EWC | {100, 100} | 10 | 84% |
| SI | {2000, 2000} | 10 | 97% |
| SI | {100, 100} | 10 | 86% |
| *A-GEM | {256, 256} | 20 | 89.1±0.14% |
| *A-GEM | {256, 256} | 10 | 92.3%** |
| *GEM | {100, 100} | 20 | 80% |
| BLLL-REG | {100, 100} | 10 | 92.2% |
| *GEM | {256, 256} | 20 | 89.5±0.48% |
| *GEM | {256, 256} | 10 | 93.1%** |
| Riemannian Walk | {256, 256} | 20 | 85.7±0.56% |
| Riemannian Walk | {256, 256} | 10 | 91.6%** |
| Progressive NNs | {256, 256} | 20 | 93.5±0.07% |
| Progressive NNs | {256, 256} | 10 | 94.6%** |

Table 1: Final average test accuracy on permuted MNIST for various methods (results taken from respective papers). A hidden layer size of $\{n_1, n_2\}$ indicates two hidden layers, the lower hidden layer having $n_1$ hidden units, the upper hidden layer having $n_2$ (followed by a softmax over the 10 MNIST classes). Methods with an asterisk (*) use some sort of episodic memory. Results marked with a double asterisk (**) were read from a graph in the source paper.

## B    MULTI-TASK TRAINING ON CIFAR-10/100

Multi-task learning, the simultaneous training of multiple tasks at the same time, is generally a complicated problem. Extra care has to be taken especially when the different tasks have different difficulties or different dataset sizes. In the mixed CIFAR-10/100 problem, the dataset for the first task is ten times larger than the other tasks. An important practical choice which affects the relative performance of the tasks is how to perform minibatch training. In particular during one epoch of training, if we draw from all tasks indiscriminately, the samples of the smaller datasets run out when only 10% of the larger dataset is seen. In this case we can either start a new epoch i.e. cut the larger dataset short or we can keep training on the larger dataset until it runs out. In practice, this choice amounts to either sacrificing the performance of the first task in favor of the other tasks or vice versa.

For the purposes of comparison to lifelong learning, we take a compromise approach. When training all tasks at the same time, we take each batch to comprise of 50 samples from task 1 and 10 samples from each of the other tasks, the total batch size adding to 140. We then start a new epoch when the the smaller datasets run out. In this way, a new epoch is started when only 50% of the large dataset of task 1 is seen. Note that the relative number of samples in each task biases the network towards one task or another, therefore picking an even higher number of samples of the first task would again lead to

sacrificing the accuracy on the other tasks. We made the choice of the 5 to 1 ratio purely as a middle ground. It is possible that other choices can lead to better overall performance.

In order to adapt the single-head model of Tab. 3 for multi-task training, we partition the neurons of the final hidden layer into 10 equal parts which form the "heads" of the 10 different tasks. We chose to train only on 10 out of the 11 tasks in order to provide a fair comparison since our continuous learning algorithm depletes the network after 10 tasks. We train using 120 epochs using Adam optimizer with learning rate 0.001 and learning rate schedule with milestones at 50 and 90 epochs and $\gamma = 0.05$. We use the same heldout validation set as continuous learning and individual learning for early stopping. We run the training 5 times and report the mean and standard deviation of the test accuracy in Fig. 5a.

## C  EXPERIMENT DETAILS

The networks used for the split CIFAR-10 and CIFAR-100 are given in tables 2 and 3.

| Layer | Chan. | Ker. | Str. | Pad. | Dropout |
|---|---|---|---|---|---|
| $32 \times 32$ input | 3 | | | | |
| Conv 1 | 32 | $3 \times 3$ | 1 | 1 | |
| Conv 2 | 32 | $3 \times 3$ | 1 | 1 | |
| MaxPool | | $3 \times 3$ | 2 | 1 | 0.25 |
| Conv 3 | 64 | $3 \times 3$ | 1 | 1 | |
| Conv 4 | 64 | $3 \times 3$ | 1 | 1 | |
| MaxPool | | $3 \times 3$ | 2 | 1 | 0.25 |
| Dense 1 | 512 | | | | 0.5 |
| Task 1: Dense | 10 | | | | |
| $\cdots$ : Dense | 10 | | | | |
| Task 6: Dense | 10 | | | | |

Table 2: Split CIFAR narrow multi-head model. Convolutional layers and Dense 1 are followed by ReLU activation.

| Layer | Chan. | Ker. | Str. | Pad. |
|---|---|---|---|---|
| $32 \times 32$ input | 3 | | | |
| Conv 1 | 128 | $3 \times 3$ | 1 | 1 |
| Conv 2 | 256 | $3 \times 3$ | 2 | 0 |
| Conv 3 | 512 | $3 \times 3$ | 1 | 0 |
| Conv 4 | 1024 | $3 \times 3$ | 2 | 1 |
| Conv 5 | 2048 | $3 \times 3$ | 1 | 0 |
| Conv 6 | 10 | $3 \times 3$ | 1 | 0 |
| AvgPool | | | | |

Table 3: Split CIFAR wide single-head model. Convolutions 1-5 are followed by BatchNorm then ReLU activation.

