# OpenReview forum: "Continual Learning via Neural Pruning"
_ICLR.cc/2020/Conference — Reject_

### Official Review · AnonReviewer3 · 2019-10-22
**Official Blind Review #3**

**Rating:** 3

**Review:**

This paper proposes a method for mitigating catastrophic forgetting in neural networks, which is an important and unsolved problem in the sequential training of multiple tasks.  The method works by (i) identifying an active subnetwork after training on one task that can perform the task to a sufficient degree of accuracy, (ii) freezing the subnetwork and pruning interfering connections to it from the rest of the network, and (iii) training the rest of the network on the next task and iterating. The contribution is a method that allows for training of sequential tasks in a fixed capacity architecture with zero catastrophic forgetting guaranteed, while still allowing for transfer between tasks via feature sharing. Additionally, it allows for controlled fine-tuning of the trade-off between accuracy and network capacity, which the authors refer to as ‘graceful forgetting’, by adjusting two hyperparameters that adjust the level of sparsity during the training of a task. The method is shown to outperform several other continual learning methods on permuted MNIST and one other method on split CIFAR-100, both standard evaluations in continual learning.
While the model demonstrates the enviable properties for a continual learning algorithm of zero forgetting and ability to transfer in a fixed network, it suffers from a significant limitation (acknowledged by the authors) that leads me to recommend it for rejection in its current incarnation:
* Due to the progressive freezing of the parameters, the network eventually reaches full capacity, making it theoretically difficult (in the multi-head setting, depending on the degree of transfer between tasks) or impossible (in the single-head setting) to learn how to perform new tasks.
* This is particularly problematic because the mechanism for 'graceful forgetting’ can only be applied on the subnetwork for a task that has just been trained on; once N tasks have been learnt, the subnetwork for task k<N cannot be altered without potentially affecting the performance on tasks k to N.
* In the paper, subnetworks are sparsified to within a fixed margin of the maximum accuracy of each task - with this procedure, you cannot predict how much capacity will be taken up by each task, and so you cannot know in advance how many tasks will fit into the network. If instead you chose to sparsify to a given subnetwork capacity per task in order to guarantee being able to fit a certain number in, then you can not control the accuracy level for each task.
* Another consequence of the method is that it is not clear how one can easily resume training on a previous task (perhaps to take advantage of transfer from subsequent tasks). You could reinstate connections from the features of subsequent tasks to the earlier task and train, but presumably these would initially interfere with the original subnetwork. Have the authors considered any ways of achieving this?
In fairness to the authors, they (a) show that the growth in utilised capacity slows as more tasks are added and (b) propose that the layers of the network could be dynamically extended as a way of overcoming a network at full capacity. For the reasons given above, however, it is not a practical fixed capacity algorithm for a lifelong learning setting, which constitutes an indefinite stream of sequential tasks, since at capacity it can suffer from sudden 'catastrophic remembering’, as opposed to the ‘graceful forgetting’ advertised in the paper. I do think that the idea has potential and that the algorithm would be *significantly* strengthened if there were a mechanism for graceful forgetting of all or selected previous tasks when the network is at capacity.

Further Comments / questions:
* The method is claimed to be the “first viable algorithm for single-head continual learning”. This statement is unspecific - what does ‘viable’ mean in this context? Online EWC [1] is a continual learning algorithm that can theoretically applied in a single-head setting - what makes it non-viable?
Minor comments not affecting review:
* Better to label the weight types in the caption of Figure 1 rather than just in the main text
* In order to see the effect of depletion more clearly, it would also be useful to see the performance of the current task in isolation rather than just the average of all previous tasks.
* Section 2, line 8: “speci[c]fic"
* The main text reference to Figure 5b compares it to the single-head MNIST usage graph - where is this?

[1] Schwarz, Jonathan, et al. "Progress & Compress: A scalable framework for continual learning." International Conference on Machine Learning. 2018.

**Experience Assessment:**

I have published one or two papers in this area.

**Review Assessment: Checking Correctness Of Derivations And Theory:**

I assessed the sensibility of the derivations and theory.

**Review Assessment: Checking Correctness Of Experiments:**

I assessed the sensibility of the experiments.

**Review Assessment: Thoroughness In Paper Reading:**

I read the paper thoroughly.

---

### Official Review · AnonReviewer2 · 2019-10-23
**Official Blind Review #2**

**Rating:** 3

**Review:**

Continual lifelong learning is very interesting and has attracted increasing attention in recent years. Generally, there are two group of methods. And, the proposed method called CLNP leverages the merits from both of them. The experimental results clearly shows the effectiveness of this method.

My main concern is that this paper actually was accepted by a Neuro AI workshop of NeurIPS and thus this submission does not comply the rules of double-blind review process. Anyway, overall, this paper is well-written and interesting.

I have more following concerns
1) As the review given by Neuro AI, the submission just compare their results to Zenke et al. 2017 for SPLIT CIFAR-10/CIFAR-100. In this version, authors does not add new experiments to compare some new methods. Actually, most concerns raised by the reviewers in Neuro AI have not been considered.
2) It is not clear how to set the hyperparameter such as threshold \theta in different layers. Most settings and criteria are given empirically and this paper seriously lacks theoretical analysis.


**Experience Assessment:**

I have read many papers in this area.

**Review Assessment: Checking Correctness Of Derivations And Theory:**

I did not assess the derivations or theory.

**Review Assessment: Checking Correctness Of Experiments:**

I did not assess the experiments.

**Review Assessment: Thoroughness In Paper Reading:**

I read the paper thoroughly.

---

### Official Review · AnonReviewer1 · 2019-10-27
**Official Blind Review #1**

**Rating:** 3

**Review:**

General:
The paper proposed neural pruning method to overcome the catastrophic forgetting. Neural pruning identifies important nodes after learning each task, and sets the values of the incoming weights to 0 to preserve the node activation values. Also, the paper propose a method for gracefully forgetting, which is certainly needed for fixed capacity network.

Pros:
1. When the network capacity does not get depleted, they showed the accuracy does not get dropped.
2. Good results of outperforming several SOTA algorithms.
3. Proposed a new method for gracefully forgetting, and achieved a good result.

Con & Questions:
1. Too many hyperparameters which will only dramatically increase with depths.
2. The method for gracefully forgetting is not very practical.
3. Depletion often happens - accuracy not dropping seems to be an overclaim.
4. I think it is almost impossible to reproduce the results of this paper.
5. What happens if all the layers have the same hyperparameters?
6. When a depletion happens for a single-headed network, then how can you learn a new task?

**Experience Assessment:**

I have published one or two papers in this area.

**Review Assessment: Checking Correctness Of Derivations And Theory:**

I assessed the sensibility of the derivations and theory.

**Review Assessment: Checking Correctness Of Experiments:**

I assessed the sensibility of the experiments.

**Review Assessment: Thoroughness In Paper Reading:**

I read the paper at least twice and used my best judgement in assessing the paper.

---

### Decision · Program_Chairs · 2019-12-19

**Decision:**

Reject

**Comment:**

There are several concerns with the brittleness and reproducibility of the proposed approach and experiments.